# SHAKEDROP REGULARIZATION

## ABSTRACT

This paper proposes a powerful regularization method named *ShakeDrop regularization*. ShakeDrop is inspired by Shake-Shake regularization that decreases error rates by disturbing learning. While Shake-Shake can be applied to only ResNeXt which has multiple branches, ShakeDrop can be applied to not only ResNeXt but also ResNet, Wide ResNet and PyramidNet in a memory efficient way. Important and interesting feature of ShakeDrop is that it strongly disturbs learning by multiplying even a negative factor to the output of a convolutional layer in the forward training pass. The effectiveness of ShakeDrop is confirmed by experiments on CIFAR-10/100 and Tiny ImageNet datasets.

## 1 INTRODUCTION

Recent advances in generic object recognition have been brought by deep neural networks. After ResNet (He et al., 2016) opened the door to very deep CNNs of over a hundred layers by introducing the residual block, its improvements such as Wide ResNet (Zagoruyko & Komodakis, 2016), PyramdNet (Han et al., 2017a;b) and ResNeXt (Xie et al., 2017) have broken the records of lowest error rates. On the other hand, in learning, they often suffer from problems such as vanishing gradients. Hence, regularization methods help to learn and boost the performance of such base network architectures. Stochastic Depth (ResDrop) (Huang et al., 2016) and Shake-Shake (Gastaldi, 2017) are known to be effective regularization methods for ResNet and its improvements. Among them, Shake-Shake applied to ResNeXt is the one achieving the lowest error rates on CIFAR-10/100 datasets (Gastaldi, 2017).

Shake-Shake, however, has following two drawbacks. (1) Shake-Shake can be applied to only multi-branch architectures (i.e., ResNeXt). (2) Shake-Shake is not memory efficient. Both drawbacks come from the same root. That is, Shake-Shake requires two branches of residual blocks to apply. If it is true, it is not difficult to conceive its solution: a similar disturbance to Shake-Shake on a single residual block. It is, however, not trivial to realize it.

The current paper addresses the problem of realizing a similar disturbance to Shake-Shake on a single residual block, and proposes a powerful regularization method, named *ShakeDrop regularization*. While the proposed ShakeDrop is inspired by Shake-Shake, the mechanism of disturbing learning is completely different. ShakeDrop disturbs learning more strongly by multiplying even a negative factor to the output of a convolutional layer in the forward training pass. In addition, a different factor from the forward pass is multiplied in the backward training pass. As a byproduct, however, learning process gets unstable. Our solution to this problem is to stabilize the learning process by employing ResDrop in a different usage from the usual. Based on experiments using various base network architectures, we reveal the condition that the proposed ShakeDrop successfully works.

## 2 EXISTING METHODS REQUIRED TO INTRODUCE THE PROPOSED METHOD

### 2.1 DEEP NETWORK ARCHITECTURES

**ResNet** (He et al., 2016) opened the door to very deep CNNs of over a hundred layers by introducing the residual block, given as

$$G(x) = x + F(x), \tag{1}$$

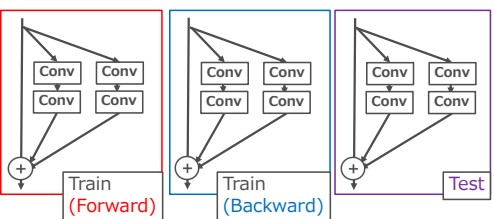

(a) ResNeXt (Xie et al., 2017), in which some processing layers omitted for conciseness.

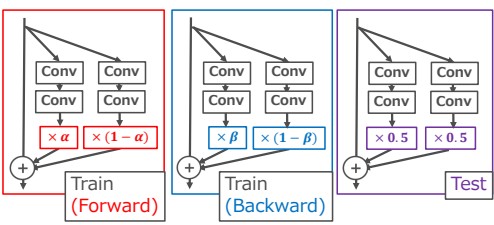

(b) Shake-Shake (ResNeXt + Shake-Shake) (Gastaldi, 2017), in which some processing layers omitted for conciseness.

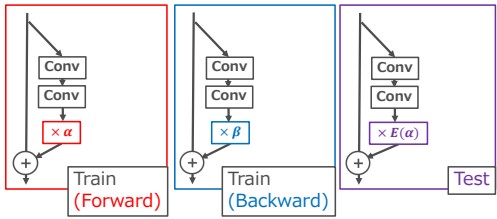

(c) PyramidNet + *"1-branch Shake."* "1-branch Shake" is an intermediate regularization method to the proposed ShakeDrop.

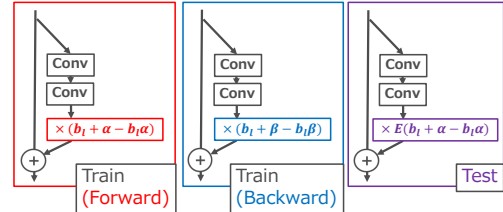

(d) PyramidNet + ShakeDrop. ShakeDrop is the proposed regularization method.

Figure 1: Network architectures.

where $x$ and $G(x)$ are the input and output of the residual block, respectively, and $F(x)$ is the output of the residual branch on the residual block.

While ResNet realized deep networks, it is imperfect. He et al. (2016) pointed out that high error rates obtained by the 1202-layer ResNet on the CIFAR datasets (Krizhevsky, 2009) is caused by overfitting. Veit et al. (2016) experimentally showed that some residual blocks whose channels greatly increase cause high error rates.

**PyramidNet** (Han et al., 2017a;b) overcame the problem of ResNet by gradually increasing channels on each residual block. It has almost same residual block as Eqn. (1). It successfully realized *deep* CNNs of up to 272 layers and achieved the lowest error rates among the vanilla residual networks on the CIFAR datasets.

## 2.2 NON-DEEP NETWORK ARCHITECTURES

**Wide ResNet** (Zagoruyko & Komodakis, 2016) improved error rates by simply increasing channels of ResNet. While it has almost same residual block as Eqn. (1), it has wider and shallower architecture than ResNet. Jastrzebski et al. (2017) experimentally showed that like PyramidNet, Wide ResNet overcomes the problem of ResNet.

**ResNeXt** (Xie et al., 2017) achieved lower error rates than Wide ResNet on almost same number of parameters. The basic architecture of ResNeXt is given as

$$G(x) = x + F_1(x) + F_2(x), \tag{2}$$

where $F_1(x)$ and $F_2(x)$ are the outputs of the residual branches as shown in Figure 1(a). The number of residual branches is not limited to 2, and the number is the most important factor to control the result.

## 2.3 REGULARIZATION METHODS

**Stochastic Depth** (Huang et al., 2016) is a regularization method which overcame problems of ResNet such as vanishing gradients. It makes the network apparently shallow in learning by dropping residual blocks stochastically selected. On the $l^{th}$ residual block from input layer, the Stochastic Depth process is given as

$$G(x) = x + b_l F(x), \tag{3}$$

where $b_l \in \{0, 1\}$ is a Bernoulli random variable with the probability of $p_l$. Huang et al. (2016) tested two rules for determining $p_l$: uniform rule and linear decay rule. The uniform rule using a constant for all $p_l$ did not work well. The linear decay rule which defines $p_l$ as

$$p_l = 1 - \frac{l}{L}(1 - p_L), \tag{4}$$

where $L$ is the number of all layers and $p_L$ is the initial parameter, worked well.

It was originally proposed for improving ResNet. It was also applied to PyramidNet; the combination of PyramidNet and Stochastic Depth was named PyramidDrop (Yamada et al., 2016).

**Shake-Shake** (Gastaldi, 2017) is a powerful regularization method for improving ResNeXt architectures. It is given as

$$G(x) = x + \alpha F_1(x) + (1 - \alpha)F_2(x), \tag{5}$$

where $\alpha$ is a random coefficient given as $\alpha \in [0, 1]$. As shown in Figure 1(b), calculation of the backward pass is disturbed by another random coefficient $\beta \in [0, 1]$ instead of $\alpha$. As a result, Shake-Shake decreased error rates than ResNeXt.

## 3 PROPOSED METHOD

### 3.1 INTERPRETATION OF SHAKE-SHAKE REGULARIZATION AND ITS DRAWBACKS

We give an intuitive interpretation of the forward pass of Shake-Shake regularization. To the best of our knowledge, it has not been given yet, while the phenomenon in the backward pass is experimentally investigated by Gastaldi (2017). As shown in Eqn. (5) (and also in Fig. 1(b)), in the forward pass, Shake-Shake interpolates the outputs of two residual branches (i.e., $F_1(x)$ and $F_2(x)$) with a random variable $\alpha$ that controls the degree of interpolation. As DeVries & Taylor (2017a) demonstrated that interpolation of two data in the feature space can synthesize reasonable augmented data, the interpolation of two residual blocks of Shake-Shake in the forward pass can be interpreted as synthesizing data. Use of a random variable $\alpha$ generates many different augmented data. On the other hand, in the backward pass, a different random variable $\beta$ is used to disturb learning to make the network learnable long time. Gastaldi (2017) demonstrated how the difference between $\alpha$ and $\beta$ affects.

The regularization mechanism of Shake-Shake relies on two or more residual branches, so that it can be applied only to 2-branch networks architectures. In addition, 2-branch network architectures consume more memory than 1-branch network architectures. One may think the number of learnable parameters of ResNeXt can be kept in 1-branch and 2-branch network architectures by controlling its cardinality and the number of channels (filters). For example, a 1-branch network (e.g., ResNeXt 1-64d) and its corresponding 2-branch network (e.g., ResNeXt 2-40d) have almost same number of learnable parameters. However, even so, it increases memory consumption due to the overhead to keep the inputs of residual blocks and so on. By comparing ResNeXt 1-64d and 2-40d, the latter requires more memory than the former by 8% in theory (for one layer) and by 11% in measured values (for 152 layers).

### 3.2 SIMILAR REGULARIZATION TO SHAKE-SHAKE ON 1-BRANCH NETWORK ARCHITECTURES

In order to realize a similar regularization to Shake-Shake on 1-branch network architectures, in the forward pass, we need a mechanism, different from interpolation, to synthesize augmented data in the feature space. Actually, DeVries & Taylor (2017a) demonstrated not only interpolation but also noise addition in the feature space works well. Hence, following Shake-Shake, we apply random perturbation, using $\alpha$, to the output of a residual branch (i.e., $F(x)$ of Eqn. (1)). In the backward pass, we can use the same way as Shake-Shake for 1-branch network architectures.

We call the regularization method mentioned above *1-branch Shake*. It is given as

$$G(x) = x + \alpha F(x), \tag{6}$$

where $\alpha$ is a coefficient that disturbs learning in the forward pass. As shown in Figure 1(c), $\beta$ is a coefficient similar to the one of Shake-Shake and used in the backward pass. The *1-branch Shake* is

expected to realize powerful generalization like Shake-Shake regularization. However, by applying it to 110-layer PyramidNet with $\alpha \in [0, 1]$ and $\beta \in [0, 1]$ following Shake-Shake, the result on the CIFAR-100 dataset was hopelessly bad (i.e., an error rate of 77.99%).

### 3.3 STABILIZING LEARNING WITH INTRODUCTION OF MECHANISM OF RESDROP

Failure of *1-branch Shake* is caused by too strong perturbation. However, weakening the perturbation would also weaken the effect of regularization. Thus, we need a trick to promote learning under strong perturbation.

Our idea is to use the mechanism of ResDrop for solving the issue. As is written in Sec.2.3, in the original form, ResDrop promotes learning by dropping some residual blocks. In other words, it makes a network apparently shallow in learning. In our situation, however, the original usage of ResDrop does not contribute because a shallower network to which *1-branch Shake* is applied would also suffer from strong perturbation. Thus, we use the mechanism of ResDrop as a probabilistic switch of two network architectures: the original network (e.g., PyramidNet) and the one to which *1-branch Shake* is applied (e.g., PyramidNet + *1-branch Shake*). By mixing up two networks, the following effects are expected.

1. When the original network (e.g., PyramidNet) is selected, learning is correctly promoted.
2. When the network with strong perturbation (e.g., PyramidNet + *1-branch Shake*) is selected, learning is disturbed.

To achieve a good performance, two networks should be in a good balance.

The proposed *ShakeDrop* is given as

$$G(x) = x + (b_l + \alpha - b_l\alpha)F(x), \tag{7}$$

where $b_l$ is a Bernoulli random variable following the linear decay rule. We call the new method "ShakeDrop." According to the value of $b_l$, Eqn. (7) is deformed as

$$G(x) = \begin{cases} x + F(x), & \text{if } b_l = 1 \\ x + \alpha F(x), & \text{otherwise (i.e., if } b_l = 0). \end{cases} \tag{8}$$

In the case that the base network is PyramidNet, ShakeDrop is equivalent to PyramidNet if $b_l = 1$, and it is equivalent to "PyramidNet + *1-branch Shake*" if $b_l = 0$. We found that the linear decay rule of ResDrop (Eqn. (4)) works well, while we also tested the uniform rule which did not work well. As shown in Figure 1(d), similar to Shake-Shake, $\beta$ is used in the backward pass instead of $\alpha$. It is noteworthy that regardless of the value of $\beta$, the weights of the network are updated. For example, let us consider the case of $\beta = 0$. In such a case, the weights of the residual blocks selected to perturbate (i.e., the layers with $b_l = 0$) are not updated. However, the inputs to their succeeding residual blocks are perturbated and the weights of the residual blocks are updated reflecting the perturbation.

## 4 EXPERIMENTS

See Sec. A for implementation details.

### 4.1 PRELIMINARY EXPERIMENTS: SEARCH FOR PARAMETER SETTINGS OF THE PROPOSED METHOD

The best parameter settings of the proposed method, ShapeDrop, are searched when it is applied to PyramidNet. They include parameter ranges of $\alpha$ and $\beta$, and the timing these parameters are updated. In the preliminary experiments, the CIFAR-100 dataset was used. PyramidNet had 110 layers, which consisted of a convolution layer, 54 additive pyramidal residual blocks and a fully connected layer. The number of channels at the last residual block was 286. Table 1 shows representative parameter ranges of $\alpha$ and $\beta$ we tested and their results. From the results, we can see how the combinations of $\alpha$ and $\beta$ affect. First of all, except PyramidDrop ($\alpha = 0, \beta = 0$), only case F ($\alpha \in [-1, 1], \beta = 0$) and case H ($\alpha \in [-1, 1], \beta \in [0, 1]$) were better than the original PyramidNet

Table 1: Average Top-1 errors (%) of "PyramidNet + ShakeDrop" with several ranges of parameters of 4 runs at the final (300th) epoch on CIFAR-100 dataset in the "Batch" level. In some settings, it is equivalent to PyramidNet and PyramidDrop.

|   | $\alpha$ | $\beta$ | Error (%) | Note |
|---|---|---|---|---|
| A | 1 | 1 | 18.01 | Equivalent to PyramidNet |
| B | 0 | 0 | 17.74 | Equivalent to PyramidDrop |
| C | $[0, 1]$ | $[-1, 1]$ | 20.61 | |
| D | $[0, 1]$ | $[0, 1]$ | 18.27 | |
| E | $[-1, 1]$ | 1 | 18.68 | |
| F | $[-1, 1]$ | 0 | 17.28 | |
| G | $[-1, 1]$ | $[-1, 1]$ | 18.26 | |
| H | $[-1, 1]$ | $[0, 1]$ | **16.22** | |

Table 2: Average Top-1 errors (%) of "PyramidNet + ShakeDrop" with different levels of 4 runs at the final (300th) epoch on CIFAR-100 dataset.

| $\alpha$ | $\beta$ | Level | Error (%) |
|---|---|---|---|
| $[-1, 1]$ | $[0, 1]$ | Batch | 16.22 |
| | | Image | 16.04 |
| | | Channel | 16.12 |
| | | Pixel | **15.78** |

(case A: $\alpha = 1, \beta = 1$). Among them, case H was the best. It is interesting to observe three cases E ($\alpha \in [-1, 1], \beta = 1$), F and H all of which take the same range of $\alpha$. Since cases F and H were better than PyramidNet, we confirm strong perturbation is effective. However, case E was even worse than PyramidNet. This implies that if strong perturbation is applied, the residual blocks selected to perturbate strongly suffer from the perturbation, so that the weights of the residual blocks should not be fully updated (i.e., $\beta = 1$ should be avoided). On the other hand, the range of $\beta$ of the best case H was in-between cases E and F. This implies that the values of $\beta$ should be well balanced.

The best strategy to update the parameters (referred as scaling coefficient) among "Batch," "Image," "Channel" and "Pixel" are explored. "Batch" means that the same scaling coefficients are used for all the images in the mini-batch for each residual block. "Image" means that the same scaling coefficients are used for each image for each residual block. "Channel" means that the same scaling coefficients are used for the each channel for each residual block. "Pixel" means that the same scaling coefficients are used for each element for each residual block. ShakeDrop was trained using the best parameters found in the experiment above, i.e, ranges $\alpha \in [-1, 1], \beta \in [0, 1]$. Table 2 shows that "Pixel" level was the best. However, it required a lot of memory. Hence, we selected "Image" level, which was the second best, considering memory efficiency.

Through the preliminary experiments, we found that the most effective parameters for $\alpha$ and $\beta$ are $\alpha \in [-1, 1]$ and $\beta \in [0, 1]$, and the effective and memory efficient level is "Image" level. We conduct the following experiments using these parameters.

## 4.2 COMPARISON WITH REGULARIZATION METHODS

The proposed ShakeDrop is compared with vanilla (without regularization), ResDrop and *1-branch Shake* in different network architectures including ResNet, PyramidNet, Wide ResNet and ResNeXt. Tables 3 and 4 show experimental results on CIFAR-100 dataset (Krizhevsky, 2009) and Tiny ImageNet dataset [1], respectively.

Table 3 shows that the proposed ShakeDrop can be applied to not only PyramidNet but also ResNet, Wide ResNet and ResNeXt. However, to successfully apply it, we found that the residual blocks have to end with batch normalization (BN) (Ioffe & Szegedy, 2015). Since without BN, the outputs of residual blocks are not in a certain range, sometimes factors $\alpha$ and $\beta$ could be too big. This seems to cause diverge in learning. With this regard, EraseReLU (Dong et al., 2017) which elimi-

---
[1] https://tiny-imagenet.herokuapp.com/

Table 3: Top-1 errors (%) at the final (300th) epoch of ResNet and its improvements to which different regularization methods are applied on CIFAR-100 dataset. Method names are followed by components of their residual blocks. For ResNet and ResNeXt, in addition to the original form, ones without ReLU unit in the end of residual blocks following EraseReLU (Dong et al., 2017) are also examined. For Wide ResNet, ones with bath normalization added in the end of residual blocks, referred as "w/ BN," are also examined. "Type A" and "Type B" of ResDrop and ShakeDrop mean that regularization unit is inserted before and after "add" unit for residual branches, respectively. "×" means learning did not converge. ∗ indicates the result is quoted from the literature.

(a) 1-residual-branch network architectures (ResNet, ResNeXt and PyramidNet)

| Methods | Regularization | Original (%) | EraseReLU (%) |
|---|---|---|---|
| **ResNet-110** <Conv-BN-ReLU-Conv-BN-add-(ReLU)> | Vanilla | 28.51 | 24.93 |
| | ResDrop | 24.09 | 22.88 |
| | 1-branch Shake | 24.18 | 23.80 |
| | ShakeDrop | × | **22.68** |
| **ResNet-164 Bottleneck** <Conv-BN-ReLU-Conv-BN-ReLU-Conv-BN-add-(ReLU)> | Vanilla | 22.00 | 21.96 |
| | ResDrop | 21.96 | 20.35 |
| | 1-branch Shake | 22.20 | 21.60 |
| | ShakeDrop | × | **19.89** |
| **ResNeXt-29 8-64d Bottleneck** <Conv-BN-ReLU-Conv-BN-ReLU-Conv-BN-add-(ReLU)> | Vanilla | 20.90 | 20.25 |
| | ResDrop | 20.66 | 20.28 |
| | 1-branch Shake | 22.70 | 24.00 |
| | ShakeDrop | × | **19.90** |
| **PyramidNet-272 $\alpha$200 Bottleneck** <BN-Conv-BN-ReLU-Conv-BN-ReLU-Conv-BN-add> | Vanilla | ∗16.35 | N/A |
| | ResDrop | 15.94 | |
| | 1-branch Shake | 71.51 | |
| | ShakeDrop | **14.90** | |

(b) 1-residual-branch network architectures (Wide ResNet)

| Methods | Regularization | Original (%) | w/ BN (%) |
|---|---|---|---|
| **Wide-ResNet-28-10k** <BN-ReLU-Conv-BN-ReLU-Conv-(BN)-add> | Vanilla | 26.49 | 24.24 |
| | ResDrop | 34.19 | 26.64 |
| | 1-branch Shake | 90.73 | 58.89 |
| | ShakeDrop | 76.87 | **19.12** |

(c) 2-residual-branch network architectures

| Methods | Regularization | Original (%) | EraseReLU (%) |
|---|---|---|---|
| **ResNeXt-164 2-1-40d Bottleneck** <Conv-BN-ReLU-Conv-BN-ReLU-Conv-BN-add-(ReLU)> | Vanilla | 23.82 | 21.75 |
| | ResDrop Type-A | 21.38 | 20.44 |
| | ResDrop Type-B | 21.34 | 20.21 |
| | Shake-Shake | 22.35 | 22.51 |
| | ShakeDrop Type-A | × | **19.98** |
| | ShakeDrop Type-B | × | **19.83** |
| **ResNeXt-29 2-4-64d Bottleneck** <Conv-BN-ReLU-Conv-BN-ReLU-Conv-BN-add-(ReLU)> | Vanilla | 21.19 | × |
| | ResDrop Type-A | 21.12 | 20.13 |
| | ResDrop Type-B | 19.27 | 19.01 |
| | Shake-Shake | 19.16 | 18.82 |
| | ShakeDrop Type-A | × | 20.07 |
| | ShakeDrop Type-B | × | **18.17** |

nates the last ReLU is a very convenient for us because in the EraseReLU versions of ResNet and ResNeXt, the residual blocks ends with BN. On the other hand, in the case of Wide ResNet, we need to intentionally add BN in the end of the residual blocks. In ResNeXt, we examined two ways, referred as "Type A" and "Type B," to apply ResDrop and ShakeDrop. They mean the regularization module is inserted before and after "add" module, respectively. As far as we examined, Type B was

Table 4: Top-1 errors (%) at the final (300th) epoch of ResNet and its improvements to which different regularization methods are applied on Tiny ImageNet dataset. Method names are followed by components of their residual blocks. For ResNet and ResNeXt, in addition to the original form, ones without ReLU unit in the end of residual blocks following EraseReLU (Dong et al., 2017) are also examined. For Wide ResNet, ones with bath normalization added in the end of residual blocks, referred as "w/ BN," are also examined. "Type A" and "Type B" of ResDrop and ShakeDrop mean that regularization unit is inserted before and after "add" unit for residual branches, respectively. "×" means learning did not converge. ∗ indicates the result is quoted from the literature.

(a) 1-residual-branch network architectures (ResNet, ResNeXt and PyramidNet)

| Methods | Regularization | Original (%) | EraseReLU (%) |
|---|---|---|---|
| **ResNet-110** 
 <Conv-BN-ReLU-Conv-BN-add-(ReLU)> | Vanilla | 42.07 | **41.24** |
| | ResDrop | 43.74 | 42.50 |
| | 1-branch Shake | 45.56 | 45.16 |
| | ShakeDrop | × | 48.92 |
| **ResNet-164 Bottleneck** 
 <Conv-BN-ReLU-Conv-BN-ReLU-Conv-BN-add-(ReLU)> | Vanilla | 38.20 | **36.52** |
| | ResDrop | 37.17 | 38.09 |
| | 1-branch Shake | 39.29 | 42.10 |
| | ShakeDrop | × | 42.80 |
| **PyramidNet-110 $\alpha$270** 
 <BN-Conv-BN-ReLU-Conv-BN-add> | Vanilla | 36.52 | N/A |
| | ResDrop | 33.97 | |
| | 1-branch Shake | 85.84 | |
| | ShakeDrop | **32.44** | |
| **PyramidNet-200 $\alpha$300 Bottleneck** 
 <BN-Conv-BN-ReLU-Conv-BN-ReLU-Conv-BN-add> | Vanilla | 32.92 | N/A |
| | ResDrop | 32.17 | |
| | 1-branch Shake | 78.12 | |
| | ShakeDrop | **31.15** | |

(b) 1-residual-branch network architectures (Wide ResNet)

| Methods | Regularization | Original (%) | w/ BN (%) |
|---|---|---|---|
| **Wide-ResNet-28-10k** 
 <BN-ReLU-Conv-BN-ReLU-Conv-(BN)-add> | Vanilla | 99.50 | 37.88 |
| | ResDrop | 99.50 | 45.80 |
| | 1-branch Shake | 98.68 | 93.62 |
| | ShakeDrop | 91.11 | **36.39** |

always better than Type A. We also confirmed that ShakeDrop Type B outperformed Shake-Shake in RexNeXt. As a conclusion, the table shows that as long as the residual blocks end with BN, the proposed ShakeDrop achieved the lowest error rates.

Table 4 shows the results on Tiny ImageNet. To conduct the experiments, it is better to use networks tuned for ImageNet. However, except PyramidNet, we could apply the same networks tuned for CIFAR-100 dataset were used due to time constraint. As a result, in ResNet, the lowest error rates were achieved by vanilla EraseReLU networks. This could be caused by use of inappropriate base networks for the task since learning ended with relatively high training loss and no regularization methods correctly worked. However, in PyramidNet and Wide ResNet, the proposed ShakeDrop achieved the lowest error rates. This confirms that the proposed ShakeDrop works on Tiny ImageNet dataset.

## 4.3 COMPARISON WITH STATE-OF-THE-ART METHODS

The proposed method was compared with state-of-the-arts on the CIFAR-10/100 datasets. State-of-the-art methods introduced some techniques that can be applied to many methods in the learning process. One is *longer learning*. While most of methods related to ResNet use 300-epoch scheduling for learning like the preliminary experiment in Sec. 4.1, Shake-Shake use 1800-epoch cosine annealing, on which the initial learning rate is annealed using a cosine function without restart (Gastaldi, 2017). Another one is *image preprocessing*. DeVries & Taylor (2017b) and Zhong et al. (2017)

showed that accuracy is improved by data augmentation which randomly fills a part of learning images. For fair comparison with these methods, we also applied them to the proposed method.

PyramidNet (including PyramidDrop and "PyramidDrop + ShakeDrop") had 272 layers, which consisted of a convolution layer, 90 additive pyramidal "bottleneck" blocks (He et al., 2016) and a fully connected layer. The number of channels at the last pyramidal residual block was 864. Table 5 shows the error rates. The proposed method, "PyramidNet + ShakeDrop," without *longer learning* and *image preprocessing*, was 3.41% on the CIFAR-10 dataset and 14.90% on the CIFAR-100 dataset. We make following comparisons with existing methods. Note that the proposed method used less numbers of parameters than rival methods in all conditions.

**(1) Fair comparison with Shake-Shake (ResNeXt + Shake-Shake) in the same condition**
"PyramidNet + ShakeDrop" combined with *longer training* achieved error rates of 2.67% (better by 0.19%) on the CIFAR-10 dataset and 13.99% (better by 1.86%) on the CIFAR-100 dataset.

**(2) Fair comparison with Cutout (ResNeXt + Shake-Shake + Cutout) in the same condition**
"PyramidNet + ShakeDrop" combined with both *longer learning* and *image preprocessing* achieved error rates of 2.31% (better by 0.25%) and 12.19% (better by 3.01%) on the CIFAR-10/100 datasets, respectively.

**(3) Comparison with the state-of-the-arts, Cutout on CIFAR-10 and Coupled Ensemble on CIFAR-100**
"PyramidNet + ShakeDrop" combined with both *longer learning* and *image preprocessing* achieved error rates of 2.31% (better by 0.25% than Cutout) and 12.19% (better by 2.85% than Coupled Ensemble) on the CIFAR-10/100 datasets, respectively.

## 5 CONCLUSION

We proposed a new stochastic regularization method ShakeDrop which can be successfully applied to ResNet and its improvements as long as the residual blocks ends with BN. Its effectiveness was confirmed through the experiments on CIFAR-10/100 and Tiny ImageNet datasets. ShakeDrop achieved the state-of-the-art performance in the CIFAR-10/100 datasets, and the best results among existing regularization methods on Tiny ImageNet dataset except some cases using inappropriate networks.

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

Table 5: Top-1 errors (%) at the final epoch (300th or 1800th) on the CIFAR-10/100 datasets. Representative methods and the proposed ShakeDrop applied to PyramidNet are compared. "Reg" represents regularization methods including ResDrop (RD), Shake-Shake (SS) and proposed ShakeDrop (SD). If "Cos" is checked, 1800-epoch cosine annealing schedule (Loshchilov & Hutter, 2016) is used following Gastaldi (2017). Otherwise, 300-epoch multi-step learning rate decay schedule is used following each method. If "Fil" is checked, the data augmentation used in Cutout (CO) (DeVries & Taylor, 2017b) or Random Erasing (RE) (Zhong et al., 2017), which randomly fills a part of learning images, is used. ∗ indicates the result is quoted from the literature. + indicates the result is quoted from Gastaldi (2017). Compared to the same condition of Cutout, the state-of-the-art, the proposed method reduced the error rate by 0.25% on CIFAR-10 and 3.01% on CIFAR-100.

| Method | Reg | Cos | Fil | Depth | #Param | CIFAR -10 (%) | CIFAR -100 (%) |
|---|---|---|---|---|---|---|---|
| Coupled Ensemble (Dutt et al., 2017) | | | | 118 | 25.7M | *2.99 | *16.18 |
| | | | | 106 | 25.1M | *2.99 | *15.68 |
| | | | | 76 | 24.6M | *2.92 | *15.76 |
| | | | | 64 | 24.9M | *3.13 | *15.95 |
| | | | | - | 50M | *2.72 | *15.13 |
| | | | | - | 75M | *2.68 | *15.04 |
| | | | | - | 100M | *2.73 | *15.05 |
| ResNeXt (Xie et al., 2017) | | ✓ | | 26 | 26.2M | +3.58 | - |
| | | | | 29 | 34.4M | - | +16.34 |
| ResNeXt + Shake-Shake (Gastaldi, 2017) | SS | ✓ | | 26 | 26.2M | *2.86 | - |
| | | | | 29 | 34.4M | - | *15.85 |
| ResNeXt + Shake-Shake + Cutout (DeVries & Taylor, 2017b) | SS | ✓ | CO | 26 | 26.2M | *2.56 | - |
| | | | | 29 | 34.4M | - | *15.20 |
| PyramidNet (Han et al., 2017b) | | | | 272 | 26.0M | *3.31 | *16.35 |
| | | ✓ | RE | 272 | 26.0M | 3.42 | 16.66 |
| PyramidDrop (Yamada et al., 2016) | RD | | | 272 | 26.0M | 3.83 | 15.94 |
| | RD | ✓ | RE | 272 | 26.0M | 2.91 | 15.48 |
| PyramdNet + ShakeDrop (Proposed) | SD | | | 272 | 26.0M | 3.41 | **14.90** |
| | SD | | RE | 272 | 26.0M | 2.89 | **13.85** |
| | SD | ✓ | | 272 | 26.0M | 2.67 | **13.99** |
| | SD | ✓ | RE | 272 | 26.0M | **2.31** | **12.19** |

Gao Huang, Yu Sun, Zhuang Liu, Daniel Sedra, and Kilian Weinberger. Deep networks with stochastic depth. *arXiv preprint arXiv:1603.09382v3*, 2016.

Sergey Ioffe and Christian Szegedy. Batch normalization: Accelerating deep network training by reducing internal covariate shift. In Francis Bach and David Blei (eds.), *Proceedings of the 32nd International Conference on Machine Learning*, volume 37 of *Proceedings of Machine Learning Research*, pp. 448–456, Lille, France, 07–09 Jul 2015. PMLR. URL http://proceedings.mlr.press/v37/ioffe15.html.

Stanisaw Jastrzebski, Devansh Arpit, Nicolas Ballas, Vikas Verma, Tong Che, and Yoshua Bengio. Residual connections encourage iterative inference. *arXiv preprint arXiv:1710.04773*, 2017.

Alex Krizhevsky. Learning multiple layers of features from tiny images. Technical report, University of Toronto, 2009.

Ilya Loshchilov and Frank Hutter. Sgdr: Stochastic gradient descent with warm restarts. *arXiv preprint arXiv:1608.03983*, 2016.

Yurii Nesterov. A method of solving a convex programming problem with convergence rate $o(1/k^2)$. *Soviet Mathematics Doklady*, 27:372–376, 1983.

David E. Rumelhart, Geoffrey E. Hinton, and Ronald J. Williams. Learning representations by back-propagating errors. *Nature*, 323:533–536, 1986.

Andreas Veit, Michael J Wilber, and Serge Belongie. Residual networks behave like ensembles of relatively shallow networks. *Advances in Neural Information Processing Systems 29*, 2016.

Saining Xie, Ross Girshick, Piotr Dollár, Zhuowen Tu, and Kaiming He. Aggregated residual transformations for deep neural networks. In *Proc. CVPR*, 2017.

Yoshihiro Yamada, Masakazu Iwamura, and Koichi Kise. Deep pyramidal residual networks with separated stochastic depth. *arXiv preprint arXiv:1612.01230*, 2016.

Sergey Zagoruyko and Nikos Komodakis. Wide residual networks. In *Proc. BMVC*, 2016.

Zhun Zhong, Liang Zheng, Guoliang Kang, Shaozi Li, and Yi Yang. Random erasing data augmentation. *arXiv preprint arXiv:1708.04896*, 2017.

## A   IMPLEMENTATION DETAILS

All implementations used in the experiments were based on the publicly available code of *fb.resnet.torch*[2]. ShakeDrop was trained using back-propagation by stochastic gradient descent with an approximate version[3] of Nesterov accelerated gradient (Nesterov, 1983) and momentum method (Rumelhart et al., 1986). 4 GPUs were used for learning acceleration; due to parallel processing, even the same layer had different values of parameters $b_l$, $\alpha$ and $\beta$ depending on GPUs.

**CIFAR-10/100 datasets**

Input images of CIFAR-10/100 datasets were processed in the following manner. An original image of $32 \times 32$ pixels was color-normalized, followed by horizontally flipped with a 50% probability. Then, it was zero-padded to be $40 \times 40$ pixels and randomly cropped to be an image of $32 \times 32$ pixels.

On PyramidNet, the initial learning late was set to 0.5 for both CIFAR-10/100 datasets following the version 2 of Han et al. (2017b), while they use 0.1 for the CIFAR-10 dataset and 0.5 for the CIFAR-100 dataset since version 3. Other than PyramidNet, the initial learning late was set to 0.1. The initial learning rate was decayed by a factor of 0.1 at $1/2$ and $3/4$ of the entire learning process (300 epochs), respectively, following Han et al. (2017b). As the filter parameters initializer, "MSRA" (He et al., 2015) was used. In addition, a weight decay of 0.0001, a momentum of 0.9, and a batch size of 128 were used with 4 GPUs. On PyramidDrop and "PyramidNet + ShakeDrop," the linear decay parameter $p_L = 0.5$ was used following Huang et al. (2016). *1-branch Shake* used $\alpha = [0, 1], \beta = [0, 1]$. ShakeDrop used parameters of $\alpha = [-1, 1], \beta = [0, 1]$ and "Image" level scaling coefficient.

**Tiny ImaneNet dataset**

Input images of Tiny ImageNet were processed in the following manner. An original image of $64 \times 64$ pixels was distorted aspect ratio and randomly cropped to be an image of $56 \times 56$ pixels. Then, brightness, contrast, and saturation of the image were randomly changed. After that, the image was color-normalized by mean and standard deviation of ImageNet, followed by horizontally flipped with a 50% probability.

Learning settings of 110-layer PyramidNet on Tiny ImageNet were almost same as on CIFAR-10/100. The initial learning late was set to 0.5. The initial learning rate was decayed by a factor of 0.1 at $1/2$ and $3/4$ of the entire learning process (300 epochs), respectively, following Han et al. (2017b). As the filter parameters initializer, "MSRA" (He et al., 2015) was used. In addition, a weight decay of 0.0001, a momentum of 0.9, and a batch size of 128 were used with 4 GPUs. Stride size of the first convolution layer was set to 2, and stride of average Pooling layer was set to 7. Learning settings of other than 200-layer PyramidNet on Tiny ImageNet were almost same as 110-layer PyramidNet. The initial learning late was set to 0.1.

Learning settings of 200-layer PyramidNet on Tiny ImageNet were almost same as on ImageNet. The initial learning late was set to 0.0125. The initial learning rate was decayed by a factor of 0.1 at $1/2$, $3/4$ and $7/8$ of the entire learning process (120 epochs), respectively, following Han et al.

---

[2]https://github.com/facebook/fb.resnet.torch

[3]Please be aware that the implementation in *sgd.lua* of Torch7 with *nesterov mode* is an approximation of the original Nesterov accelerated gradient and momentum method. See https://github.com/torch/optim/issues/27.

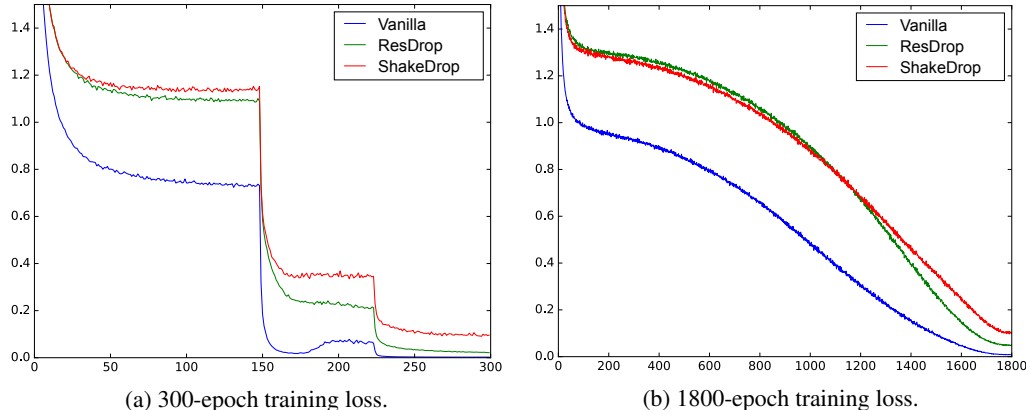

(a) 300-epoch training loss.

(b) 1800-epoch training loss.

Figure 2: Change of training loss of PyramidNet with no regularization method (vanilla), ResDrop and ShakeDrop.

(2017b). As the filter parameters initializer, "MSRA" (He et al., 2015) was used. In addition, a weight decay of 0.0001, a momentum of 0.9, and a batch size of 32 were used with 4 GPUs. Stride size of the first convolution layer was set to 1, and Stride of Max Pooling layer was set to 1.

## B    DISCUSSION

In order to find out why the proposed method achieved the low error rates, we investigate what kind of difference exists with the conventional methods, particularly with PyramidNet and PyramidDrop.

### B.1    TRAINING LOSS

Fig. 2 shows change of training loss. Fig. 2(a) is of the 300-epoch multi-step learning rate decay schedule with 110-layer networks. Fig. 2(b) is of 1800-epoch cosine annealing schedule with 272-layer networks. ResDrop and ShakeDrop, which are probabilistic regularization methods, decreased the training loss more slowly than vanilla PyramidNet in both figures. Even at the final epoch, the loss of ResDrop and ShakeDrop are much larger than 0 in contrast to vanilla PyramidNet whose loss is close to 0.

### B.2    MEAN AND VARIANCE OF GRADIENTS

In addition to the training loss, we examined the mean and variance of gradients on each iteration for each regularization method. Under the same conditions as the preliminary experiment in Sec. 4.1, we compared PyramidNet with no regularization method (vanilla), ResDrop and ShakeDrop in the "Batch" level. We focused on 3 residual blocks out of 54 residual blocks and named "first," "middle" and "final." They mean the first, 27th and final residual blocks, respectively. For each, we checked the 2nd convolutional layer. Figs. 3 and 4 show change of mean and variance of gradients, respectively. Although ResDrop and ShakeDrop were similar in Fig. 2, ShakeDrop took much larger values than ResDrop in Figs. 3 and 4. Particularly, the difference was significant in "first," and also large in "middle." They would indicate that relatively intensive training lasts in ShakeDrop.

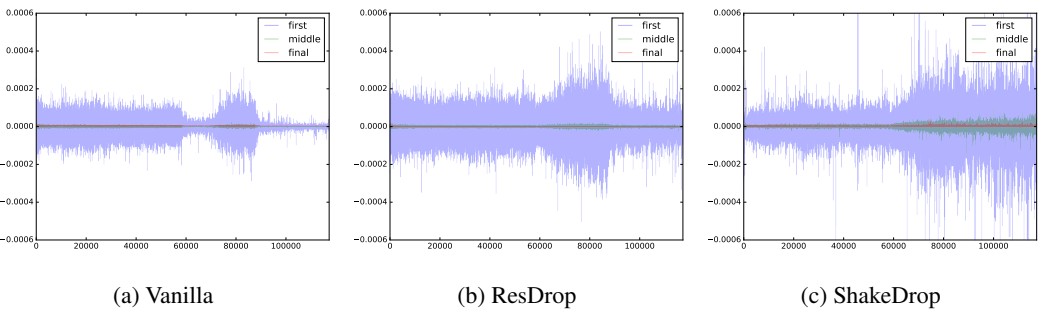

(a) Vanilla        (b) ResDrop        (c) ShakeDrop

Figure 3: The average of the gradients on each iteration in PyramidNet.

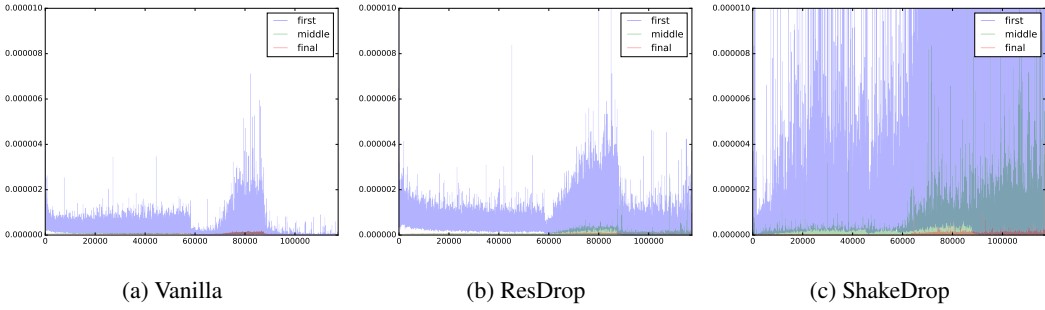

(a) Vanilla        (b) ResDrop        (c) ShakeDrop

Figure 4: The variance of the gradients on each iteration in PyramidNet.

