# OpenReview forum: "ShakeDrop regularization"
_ICLR.cc/2018/Conference — Reject_

### Official Review · AnonReviewer3 · 2017-11-24
**Compelling experimental results, analysis not totally clear**

**Rating:** 5
**Confidence:** 2

**Review:**

This paper proposes a regularization technique for deep residual networks.  It is inspired by regularization techniques which disturb the training by applying multiplicative factors to the convolutional layer outputs e.g  Shake-Shake (Gastaldi '17) and PyramidDrop (Yamada '16).  The proposed approach samples a Bernoulli variable randomly to either follow the standard variant of Pyramid net, or applies a variant of shake-shake to pyramid net.

+ Experimental results on CIFAR-10 and CIFAR-100 well-exceed exceed the existing "vanilla" techniques + regularizers.
- Clarity: some statements are not clear / not substantiated e.g. how does the proposed method overcome the memory problem that shake-shake has?  There are some minor issues wrt presentation, e.g. grammatical correctness of sentences, consistent usage of references, which can be fixed with more careful proofreading.
- Quality: even though the experimental results are compelling, the paper lacks thorough analysis in understanding the effects of the regularizer.  The two experiments looks at (1) the training error, which the paper openly states does not explain why the proposed regularization works and (2) variance of the gradients throughout learning; the larger variance of gradients is speculated to be the cause, but this is almost expected, given that the method is designed to allow larger fluctuations and perturbations during training.

---

> ### Author Response · Authors · 2018-01-03
> **Response to review comment**
>
> Thank you very much for your review comments.
>
> Here are our responses to your comments.
>
> - Clarity: Based on your comments, we substantially improved the paper. In the revised paper, we more clearly state the motivation including the memory issue, the problem we tackled and its difficulty, idea to solve the problem, interpretation of Shake-Shake regularization to derive the proposed regularization method, and experimental results including the condition of base network architectures to apply the proposed ShakeDrop into greater details.
>
> - Quality: We found that the results in question do not have so informative to explain the phenomenon. So, in the revised paper, we added further consideration regarding the range parameters (alpha and beta) and the condition of base network architectures to apply the proposed ShakeDrop.

---

### Official Review · AnonReviewer1 · 2017-11-30

**Rating:** 4
**Confidence:** 4

**Review:**

The paper proposes a new form of regularization that is an extension of "Shake-Shake" regularization (Gastaldi, 2017). The original "shake-shake" proposes using two residual paths adding to the same output (so x + F_1(x) + F_2(x)), and during training, considering different randomly selected convex combinations of the two paths (while using an equally weighted combination at test time). However, this paper contends that this requires additional memory, and attempt to achieve similar regularization with a single path. To do so, they train a network with a single residual path, where the residual is included without attenuation in some cases with some fixed probability, and  attenuated randomly (or even inverted) in others. The paper contends that this achieves superior performance than choosing simply a random attenuation for every sample (although, this can be seen as choosing an attenuation under a distribution with some fixed probability mass at 1). Experiments show improved generalization on CIFAR-10 and CIFAR-100.

I don't think the paper contains sufficiently novel elements to be accepted as a conference track paper at ICLR. While it is interesting that this works well (especially the "negative" weight on the residual), the proposed method is fundamentally a combination of prior work: dropout and "shake-shake" regularization. Moreover, the evaluation is somewhat limited---essentially, I feel there isn't conclusive proof that "shake-drop" is a generically useful regularization technique. For one, the method is evaluated only on small toy-datasets: CIFAR-10 and CIFAR-100. I think at the very least, evaluation on Imagenet is necessary. The proposed regularization is applied only to the "PyramidNet" architecture---which begs the question of whether the proposed regularization is useful only for this specific network architecture. It would have been more useful to see results with and without "shake-drop" on different architectures (the point being to show a consistent improvement with this regularization, rather than achieving 'state of the art' on CIFAR-10). Moreover, it would be interesting to see if the hyperparameter comparison shown in Tables 1 and 2 remained consistent across architectures.

---

> ### Author Response · Authors · 2018-01-03
> **Response to review comment**
>
> We appreciate your valuable feedback.
>
> We found factual errors in your comment.
> (1) The proposed method is not a combination of dropout and shake-shake.
> We agree that a combination of “dropout” and shake-shake regularization is trivial. But, it is not what we did. In the proposed ShakeDrop, we used ResDrop in a different usage from the usual. ResDrop is not used for dropping some layers as in the original paper. Instead, the mechanism of ResDrop is used as a probabilistic switch of two networks. We show in the paper that such usage of ResDrop contributes to stabilize a network hard to train. This is novel and must be informative in the community. Furthermore, in the revised paper, we present how the problem is not trivial and greater details about our interesting findings.
> (2) While this is not a clear error, we are afraid that we cannot get which method you mean by saying “choosing simply a random attenuation for every sample” of the following sentence: “The paper contends that this achieves superior performance than choosing simply a random attenuation for every sample (although, this can be seen as choosing an attenuation under a distribution with some fixed probability mass at 1).“
>
> We found your comments are reasonable. So, based on your comments, we extended experiments in two aspects in the revised paper.
> (1) The proposed ShakeDrop has been successfully applied to ResNet (EraseReLU version), Wide ResNet (with batch normalization added in the end of residual blocks) and ResNeXt (EraseReLU version) since we found that batch normalization is required to be at the end of residual blocks.
> (2) In addition to CIFAR-10/100 datasets, we confirm the effectiveness of the proposed ShakeDrop through experiments on Tiny ImageNet dataset. Unfortunately, experiments on ImageNet dataset was not possible in time with our computational resources.
>
> We hope you find our revised paper is valuable.

---

> > ### Comment · AnonReviewer1 · 2018-01-12
> > **post rebuttal**
> >
> > ---
> > Regd. the 'factual errors':
> >
> > 1. My original review said "the proposed method is *fundamentally* a combination of prior work" --- in that the underlying ideas had been introduced before in prior work (dropout & shake shake), not that the proposed method involved literally applying a combination of dropout and shake shake. As the paper notes, shake shake and dropout come out to be special cases of the proposed method (if one chose b_l to always be 0 with probability 1, or if one chose alpha to be = 0). My point was that the framework can be seen as a combination of the underlying mechanisms in both.
> >
> > 2. Attenuation = weighting by alpha. This would be shake shake (where the residual path is just weighted by alpha that is uniformly distributed in some range, rather than the proposed method where the attenuation/weighting is only applied if the randomly sampled bernoulli variable is 1).
> > --
> >
> > The proposed method is easy to understand, and the new experiments are certainly welcome (although, I think the evaluation remains unconvincing without experiments on a large scale task such as Imagenet). However, I just don't think the contribution is novel enough to be accepted as a paper at ICLR. Therefore, I'm inclined to stay with my original evaluation/score.

---

### Official Review · AnonReviewer2 · 2017-12-03

**Rating:** 4
**Confidence:** 3

**Review:**

The paper proposes ShakeDrop regularization, which is essentially a combination of the PyramidDrop and Shake-Shake regularization. The procedure consists of essentially weighting the residual branch with a random weight, in the style of Shake-Shake, where the weight is sampled from a mixture of uniform distribution in [-1, 1] and delta at 1, such that the mixture of those two distributions varies linearly with layer depth, in the style of PyramidDrop. In the style of Shake-Shake, a different random weight (in [0, 1]) is used for the backward pass. The most surprising part is that that the forward weight can be negative thus inverting the output of a convolution. Apparently the goal is to "disturb" the training, and the procedure yields state-of-the-art results on CIFAR-10/100.

Positives:

- Results: state-of-the-art on CIFAR-10/100

Negatives:

1. No real motivation on why should this work. I guess the motivation is the mixture of PyramidDrop and Shake-Shake motivations, but the main surprising part (forward weight can be negative) is not motivated at all. There is a tiny bit of discussion at the very end, section 4.4, where authors examine the training loss (showing it's non-zero so less overfitting) and mean/variance of gradients (increased). However, this doesn't really satisfy me - it is clear that more disturbance will cause this behaviour, but that doesn't mean any disturbance is good, e.g. if I always apply the negative weight and make my model weights go in the wrong direction, I'm pretty sure training loss and gradients will be even larger, but it's a bad idea to do.

2. I'm concerned with the "weird trick that happens to work on CIFAR" line of work (not saying that this paper is the only offender) - are these methods actually useful and generalizable to other problems, or are we overfitting on CIFAR and creating MNIST v2.0 ? It would be nice to demonstrate that this regularization works in at least one more problem, maybe ImageNet, though maybe regularization is not needed there but just find one more dataset that needs regularization and test this on that.

3. The paper doesn't explain well what is the problem with Shake-Shake and memory. I see that the author of Shake-Shake has made a comment on this and that makes a lot of sense, i.e. there is no memory issue, just because there are 2x branches doesn't mean shake-shake needs 2x memory as it can use less capacity=memory to achieve the same performance. So it seems the main premise of the paper - "let's apply Shake-Shake to deeper models but we need to come up with a modified method because Shake-Shake cannot be applied due to memory problems" - seems wrong.

4. The writing quality is quite bad, it is very hard to understand what authors mean in parts of the text. E.g. at two places "it has almost the same residual block as Eqn. (1)" - how is it "almost"? Below equation 5, it is never specified that alpha and beta are sampled uniformly(?) from those ranges, one could think that alpha and beta are fixed constants that take a specific value that is in that range. There are also various grammatical errors such as "is expected to be powerful but slight memory overhead" or "which is introduced essence", etc.

Smaller comments:
- Isn't it surprising that alpha in [-1, 1] and beta in [0, 1] works well, but alpha in [0, 1] and beta in [-1, 1] works much worse? The two important cases, (alpha negative, beta positive) and (alpha positive, beta negative), seem to me like they are conceptually very similar.
- End of section 4.1, should it be b_l as p_L is a constant and b_l is what is sampled?
- I don't like that exactly the same text is repeated 3 times (abstract, end of intro, end of 1.1) and in very short distance from each other - repeating the same words 3 times doesn't make the reader understand it better, slight rephrasing is much more beneficial.

Overall:
Good to know that this method sets the new state of the art on CIFAR-10/100, so as such it should be of interest to the community to be available online (arXiv). But with fairly little novelty (is a combination of 2 methods), very little insights of why this should work at all (especially the negative scaling coefficient which is the only extra thing that one learns from this paper, since the rest is a combination of PyramidDrop and Shake-Shake), no idea on whether the method would work outside of the CIFAR-world, and bad quality of the text - I don't think the manuscript is sufficiently good for ICLR.

---

> ### Author Response · Authors · 2018-01-03
> **Response to review comment**
>
> We appreciate your detailed review comments. Based on your feedback, we substantially improved the paper.
> We believe that our contribution is not limited to achieving the state of the art of CIFAR-10/100 but also providing interesting insight to the community.
>
> First of all, we would like to point out a factual error which might be caused by our paper quality.
> Regarding the novelty, we understand that the reviewer regards the proposed ShakeDrop as a simple combination of two methods (ResDrop and Shake-Shake). Though apparently it could be seen like that, it is not true. To clarify it, we enumerate what we contribute for proposing ShakeDrop.
> (1) As Shake-Shake does not work on a single residual branch, we proposed a new regularization method working on a single residual branch (used in the intermediate method (“1-branch Shake”; previously we called it PyramidShake)). While it is inspired by Shake-Shake, it is completely different one.
> (2) We used ResDrop in a different usage from the usual. In the original paper, ResDrop is used for dropping some layers. Instead, we used it as a switch of two networks. We demonstrated in the paper that such usage of ResDrop contributes to stabilize a network hard to train like “1-branch Shake.”
>
> The following are responses to negative aspects of the paper in your comments.
>
> 1. Motivation
> At the time of the initial submission, our motivation was to propose an effective and memory efficient regularization method applicable to PyramidNet because PyramidNet was the best network architecture on CIFAR-10/100 datasets.
> After reading review comments, we slightly updated our motivation. That is, in the revised paper, the target network architectures are not only PyramidNet but also ResNet, Wide ResNet and ResNext.
> Though the negative forward weight could be sensational, it is not our central contribution (we explained in the revised manuscript into greater detail). We added consideration on why the negative forward weight works well on the proposed ShakeDrop.
>
> 2. Experiments on different datasets
> It is understandable reaction to request experiments on different datasets. We added experiments on Tiny ImageNet dataset. While we understand experiments on ImageNet dataset are better, we found it is not possible to complete them in time with our computational resources.
>
> 3. Memory issue of Shake-Shake
> It seems our explanation was not appropriate. As we responded to the post by the author of Shake-Shake, out intention is not taking into account only learnable parameters but the total memory consumption.
>
> Anyway, out intention is as follows (as is written in the revised paper). Shake-Shake is designed to take a weighted sum of outputs of two residual branches. So, it requires at least two branches in a layer to apply. Due to this, it can be applied only to ResNeXt (having multiple branches) and it requires more memory to make the network deep than networks with a single residual branch in a layer.
>
> 4. Writing quality
> We are very sorry about it. We tried our best to correct such issues.
>
> Respond to “Smaller comments”
> - (alpha negative, beta positive) and (alpha positive, beta negative) are conceptually same?
> It is a reaction we expected. The answer is no. We tried our best to explain this in the revised paper.
>
> - End of section 4.1, should it be b_l as p_L is a constant and b_l is what is sampled?
> Thanks for pointing this out. It is a typo. It should have been b_l.
>
> - Exactly the same text is repeated 3 times
> We are sorry for this. It is also solved in the revised paper.

---

### Public Comment · (anonymous) · 2017-11-08
**Implementation Clarification**

Quick question, do you sample one bernoulli variable b_l during the forward pass and then save it and use it again on the backward pass, or are the bernoulli variables independently sampled on both the forward and backward passes? Thanks.

Additionally, there would appear to be an error in Figure 2(d) for the backward pass, where it has the equation as (b_l + \beta - b_l), instead of (b_l + \beta - (b_l * \beta)).

---

> ### Author Response · Authors · 2017-11-09
> **Reply to "Implementation Clarification"**
>
> Thank you very much for your inquiry.
> The answer of your question is the former.
> That is, we sample b_l on the forward pass and reuse it on the backward pass.
>
> Regarding Fig. 2(d), yes, what you have pointed out is correct.
> We will revise it on a later version.
> Thank you very much for pointing out the mistake.

---

### Public Comment · ~Yauheni_Selivonchyk1 · 2017-11-21
**Memory issue clarification**

Can you please explain in a few more words what is the memory issue with Shake-Shake networks? Namely, in which way Shake-Shake network is different in memory consumption from, say, a ResNet. And how does Shake-Drop addresses/solves this problem. Thank you.

---

> ### Author Response · Authors · 2017-11-27
> **Reply to "Memory issue clarification"**
>
> Roughly speaking, Shake-Shake requires as twice the amount of memory as ResNet on a residual block due to twice the number of residual branches. ShakeDrop can solve the issue by using a single residual branch (this corresponds to PyramidShake). Since PyramidShake is unstable in learning, we combined it with ResDrop to stabilize it.

---

> > ### Public Comment · ~Xavier_Gastaldi1 · 2017-12-02
> > **There is no memory issue**
> >
> > If I understand correctly, the authors assume that, for Shake-Shake regularization to bring an improvement, you have to keep the same number of filters, add another branch and then apply Shake-Shake regularization.
> >
> > While I can understand why the authors would assume that, the tests below paint a different story. The models are the same as in the Shake-Shake regularization paper. They were run once and Shake-Shake regularization was not applied  (i.e. Even-Even-Batch in the paper):
> >
> > A. 26 layers, 1 residual banch, 32 filters, 1.47M params: 4.69% test error
> > B. 26 layers, 2 residual banches, 22 filters, 1.37M params: 4.65% test error
> > C. 26 layers, 1 residual banch, 22 filters, 0.696M params: 5.35% test error
> > D. 26 layers, 2 residual banches, 16 filters, 0.736M params: 5.11% test error
> > E. 26 layers, 1 residual banch, 16 filters, 0.369M params: 5.98% test error
> > F. 26 layers, 2 residual banches, 12 filters, 0.416M params: 5.59% test error
> > G. 26 layers, 1 residual banch, 12 filters, 0.209M params: 7.12% test error
> > H. 26 layers, 2 residual banches, 8 filters, 0.186M params: 7.24% test error
> >
> > [B,C],[D,E],[F,G] have the same number of filters per residual branch.
> > [A,B],[C,D],[E,F],[G,H] have roughly the same capacity.
> >
> > If the author's claim was correct then we would observe the same error rates for [B,C],[D,E],[F,G]. What we see in practice is that [A,B],[C,D],[E,F],[G,H] have roughly the same error rates. This means that what is important is the total capacity of the model not the number of filters per residual branch.
> >
> > To apply Shake-Shake regularization correctly, you should add a second branch, reduce the number of filters to get back to the capacity of your initial 1 branch model and then apply Shake-Shake regularization. Following this procedure does not lead to a memory issue.

---

> > > ### Author Response · Authors · 2017-12-06
> > > **That's not what we intend**
> > >
> > > We are afraid that you don't correctly understand what we claim.
> > > We know that we can keep the number of parameters by adjusting Cardinality and baseWidth on ResNeXt.
> > > But, we don't talk about the number of parameters solely.
> > > Instead, we talk about memory consumption.
> > > Amount of required memory depends on not only the number of learnable parameters but also other factors (such as the input of each layer for calculation of gradients on the backward pass). They cause the overhead we pointed out.
> > >
> > > We found that our paper in the current form is not correctly understood by readers. So, we are improving it. Please wait for a while for the revised version.
> > >
> > > Anyway, thanks for your interest to our paper.

---

### Author Response · Authors · 2018-01-03
**Revised paper is uploaded**

We substantially improved the paper. In the revised paper, we updated as follows.
- Added experiments on different network architectures; not only PyramidNet, but also ResNet, WRN and ResNeXt
- Added experiments on a new dataset “Tiny ImageNet”
- Added consideration about parameters (alpha and beta)
- Revised introduction to fit the updated; the paper does not focus only on PyramidNet anymore
- The intermediate method specially focusing on PyramidNet, named PyramidShake, was deleted. Instead, we named an intermediate regularization method “1-branch Shake”
- Improved paper writing quality

---

> ### Author Response · Authors · 2018-01-05
> **We will revise the paper again**
>
> We found some errors that should be corrected. Now we are revising it and will upload a further revised version of the paper today.

---

### Author Response · Authors · 2018-01-05
**A new version of the paper is available**

We have cleaned up "bugs" in the revised paper.

---

### Decision · Program_Chairs · 2018-01-29
**ICLR 2018 Conference Acceptance Decision**

**Decision:**

Reject

**Comment:**

The paper proposes a regularisation technique based on Shake-Shake which leads to the state of the art performance on the CIFAR-10 and CIFAR-100 dataset. Despite good results on CIFAR, the novelty of the method is low, justification for the method is not provided, and the impact of the method on tasks beyond CIFAR classification is unclear.